# Pathogenic Puppetry: Manipulation of the Host Actin Cytoskeleton by *Chlamydia trachomatis*

**DOI:** 10.3390/ijms21010090

**Published:** 2019-12-21

**Authors:** Liam Caven, Rey A. Carabeo

**Affiliations:** 1School of Molecular Biosciences, Washington State University, Pullman, WA 99164, USA; liam.caven@wsu.edu; 2Department of Pathology and Microbiology, University of Nebraska Medical Center, Omaha, NE 68198-5900, USA

**Keywords:** actin cytoskeleton, chlamydia, bacterial pathogenesis

## Abstract

The actin cytoskeleton is crucially important to maintenance of the cellular structure, cell motility, and endocytosis. Accordingly, bacterial pathogens often co-opt the actin-restructuring machinery of host cells to access or create a favorable environment for their own replication. The obligate intracellular organism *Chlamydia trachomatis* and related species exemplify this dynamic: by inducing actin polymerization at the site of pathogen-host attachment, *Chlamydiae* induce their own uptake by the typically non-phagocytic epithelium they infect. The interaction of chlamydial adhesins with host surface receptors has been implicated in this effect, as has the activity of the chlamydial effector TarP (translocated actin recruitment protein). Following invasion, *C. trachomatis* dynamically assembles and maintains an actin-rich cage around the pathogen’s membrane-bound replicative niche, known as the chlamydial inclusion. Through further induction of actin polymerization and modulation of the actin-crosslinking protein myosin II, *C. trachomatis* promotes egress from the host via extrusion of the inclusion. In this review, we present the experimental findings that can inform our understanding of actin-dependent chlamydial pathogenesis, discuss lingering questions, and identify potential avenues of future study.

## 1. Introduction

The cytoskeleton is a highly dynamic structural framework composed of actin, microtubules, intermediate filaments, and septins. Restructuring of the cytoskeleton’s actin component is critical for a variety of cellular processes, including endocytosis, motility, nutrient acquisition, and mitosis. It should therefore be unsurprising that modulation of the actin structure and function is a common theme amongst intracellular and extracellular pathogenic bacteria: by manipulating the cytoskeleton, these organisms can induce their own uptake by host cells, scavenge nutrients from host organelles, and ultimately establish a niche that facilitates their own replication.

The structure of the actin cytoskeleton is dynamically regulated by the recruitment of actin-polymerizing (or depolymerizing) factors, which consequently alter the balance of monomeric/globular actin (G-actin) and filamentous actin (F-actin) in the cytosol. Nucleation of a new actin filament requires the formation of a thermodynamically unfavorable actin trimer; to bypass this requirement, the Arp2/3 (actin-related protein) complex recruits a single G-actin monomer alongside two structural analogs to form a site of nucleation [1,2,3]. Subsequent F-actin branching by Arp2/3 is regulated by nucleation promotion factors (NPFs), such as N-WASP, SCAR/WAVE, and WASH [4,5]. The binding of NPFs to G-actin (Type I) or F-actin (Type II) facilitates conformational changes in Arp2/3 that enhance the complex’s F-actin branching activity. NPF activation is in turn regulated by the Rho family GTPases RhoA-C, Rac1, and Cdc42—all of which are targets for modulation by pathogens seeking to restructure actin and thereby facilitate pathogenesis [6,7].

The manipulation of host actin can promote a wide variety of beneficial outcomes for the pathogen. *Salmonella* spp. translocate the effectors SopE and SopE2 into host cells—these guanine exchange factor (GEF) mimics enhance the activity of Rac1 and Cdc42, creating localized concentrations of F-actin at the apical surface of mucosal epithelia [8,9,10,11]. The result is extensive ruffling of the plasma membrane at the site of *Salmonella* attachment, leading to internalization of the pathogen via micropinocytosis [8,12,13,14]. Upon internalization and escape into the host cytosol, the Gram-positive intracellular pathogen *Listeria monocytogenes* induces the polymerization of actin on the bacterial surface through the activity of ActA, a surface protein functionally analogous to the nucleation promotion factor WASP [15]. ActA recruits an Arp2/3 complex to the bacterial pole, resulting in branched actin polymerization producing a comet-shaped structure that propels *Listeria* across the cytosol and into adjacent uninfected cells [16,17,18,19,20].

Indeed, this dynamic can be observed even in non-invasive bacterial pathogens. Enteropathogenic and enterohemorrhagic *E. coli* (EPEC/EHEC) induce the formation of distinctive, actin-rich pedestals that facilitate their attachment to gastric epithelia. The virulence factor Tir is responsible for this effect: upon delivery into host cells by the *E. coli* type III secretion system (T3SS), Tir is incorporated into the plasma membrane, promoting EPEC/EHEC attachment via binding to the bacterial adhesin intimin [21,22]. This clusters Tir at the site of attachment, inducing the phosphorylation of Tir’s cytosolic domain by host kinases and the subsequent recruitment of Nck [23,24]. Nck is an adaptor protein that binds and activates N-WASP—consequently, the downstream effect of the Tir/Nck interaction is the recruitment of N-WASP and Arp2/3 complexes at sites of EPEC/EHEC attachment [25]. The resulting polymerization of branched actin produces pedestal formation, effacing the microvillar structure of the gastric mucosa and facilitating EPEC/EHEC colonization of the gastrointestinal tract [26,27].

The Gram-negative *Chlamydia* spp. constitute a valuable model for the study of actin modulation by bacterial pathogens. As obligate intracellular parasites, *Chlamydia trachomatis* and related species restructure actin in a variety of ways, to facilitate host invasion, maintain their replicative niche, and egress from host epithelial cells. Multiple *C. trachomatis* serovars have been isolated with distinct tissue tropism in the host: serovars A–C infect the conjunctival epithelium (producing the species’ eponymous fibrotic trachoma), whereas serovars D–K and L1–L3 colonize the urogenital and anogenital tracts, respectively [28,29]. This extensive tissue tropism demonstrates a capability to modulate actin in multiple epithelial cell types, further borne out by the observation of pathogen-directed actin rearrangement by the respiratory pathogen *C. pneumoniae* [30,31], as well as the mouse- and guinea pig-infecting *C. muridarum* and *C. caviae* [32,33,34]. The study of chlamydial pathogenesis thus has the potential to reveal striking insight into both the pathogenic and steady-state regulation of actin in the host. In this review, we will summarize the field’s current understanding of actin modulation by *Chlamydiae* both during and after host invasion, as well as discuss potential avenues of further research. 

## 2. A Multilayered Assault: *Chlamydia* Redistributes the Actin Cytoskeleton to Invade Host Cells

The initial study of chlamydial invasion emphasized the importance of actin recruitment at sites where the infectious form of *Chlamydiae* (the elementary body, or EB) adheres to the host cell surface [35,36]. This early observation of in vitro infections occurred concomitant with the formation of microvillar structures that surround (and presumably internalize) invading *Chlamydia* [36,37]. The pharmacological disruption of F-actin (via cytochalasin D) or sequestration of G-actin (via latrunculin B) substantially inhibits chlamydial invasion and microvillar formation, suggesting that actin polymerization (not simply recruitment) is critical to fostering entry of the pathogen [35,36,37]. Furthermore, live-cell imaging of invasion events after cytochalasin D washout revealed the selective reestablishment of microvilli sites of EB attachment, indicating that this phenomenon is highly specific and pathogen-directed [36]. Clearly, the attachment of *Chlamydia* to the host cell surface is a critical first step to invasion, but how does EB attachment lead to actin rearrangement in the host? Extensive study of chlamydial invasion over the past two decades has suggested two, likely complementary, mechanisms: first, the engagement of host receptors by chlamydial adhesins facilitates the induction of actin polymerization indirectly, and second, bacterial effectors delivered by *Chlamydia* into the host remodel the actin cytoskeleton directly. The findings underlying these hypotheses are reviewed in the following sections.

### 2.1. Actin Modulation during Transient Chlamydial Adhesion

An early observation of *C. trachomatis* infections in vitro was that host attachment appeared to occur in two distinct stages: a reversible and temperature-insensitive interaction, followed by irreversible adhesion that requires physiological temperature [38,39,40]. Initial, transient EB attachment has since been shown to be an electrostatic interaction mediated by the glycosaminoglycan (GAG) heparan sulfate [41,42]. Heparan sulfate is demonstrably required for invasion by chlamydial serovars L1–L3, with other serovars exhibiting varying levels of GAG requirement [38,43]. Produced by a variety of cell types, heparan sulfate exhibits a strong negative charge that permits its binding to bacterial surface proteins; indeed, an early proposed mechanism for EB attachment posited the formation of a tripartite “molecular bridge” between heparan sulfate and host/bacterial GAG-binding proteins [44,45,46,47]. The bacterial surface protein OmcB is considered the primary GAG-binding protein in serovars whose adhesion is dependent on heparan sulfate [48,49,50]. OmcB has been shown to bind the alternative GAG heparin in vitro, and exhibits a variant amino acid sequence in the GAG-independent serovar E [50]. The chlamydial major outer membrane protein (MOMP) of *C. muridarum* has also been shown to bind GAGs to mediate invasion: treating host cells with recombinant MOMP or OmcB markedly reduces EB-host binding, as does treatment with monoclonal antibodies against *C. muridarum* MOMP [51]. 

The host component of this GAG-mediated bridge between EB and the host has remained elusive; however, recent work suggests an alternative, GAG-independent OmcB/MOMP attachment mechanism. Mounting evidence indicates that both proteins are post-translationally modified via glycosylation, as OmcB/MOMP recovered from EBs shows evidence of modification by N-linked high-mannose oligosaccharides [52,53,54]. Furthermore, both proteins exhibit reactivity to *Erythrina crista-galli* lectin, which binds to sites of N-acetyllactosamine glycosylation [54]. It has been further shown that glycosylated OmcB and MOMP are recognized by the protein galectin-1 (Gal1), which is both secreted by the host and bound on the plasma membrane to surface receptors [54]. Given that galectin-1 can demonstrably bridge EBs to host Gal1 receptors [54], both its secreted and membrane-associated state could conceivably facilitate EB-host binding. However, the latter state presents an intriguing avenue for the chlamydial induction of actin recruitment and invasion: in neurons, surface-associated Gal1 has been shown to initiate clathrin-independent endocytosis upon ligand binding, and Gal1 internalization is associated with F-actin polymerization during axonal growth [55].

A clathrin-independent mechanism of chlamydial invasion is of particular interest, given that clathrin’s role in chlamydial invasion is somewhat controversial. In 1999, Boleti et al. assessed the importance of clathrin-mediated endocytosis to invasion via the expression of a dominant negative mutant of Eps15 [56]. Eps15 interacts with the clathrin adaptor protein AP-2 at clathrin-coated pits [57,58]; ablating this interaction thus inhibits clathrin-mediated endocytosis [59,60]. Importantly, the invasion of both *C. trachomatis* serovar L2 and *C. caviae* was unaffected by the expression of dominant negative Eps15, suggesting that clathrin is largely dispensable for chlamydial invasion [56]. However, a more recent study involving RNAi-mediated clathrin knockdown during serovar L2 infection presented a small (but statistically significant) invasion defect [61]. A likely explanation for these conflicting results is that *Chlamydia* employs multiple, functionally redundant mechanisms of invasion, each of varying clathrin dependence. The interaction of OmcB and MOMP with surface-associated Gal1 may enhance clathrin-independent invasion, either by other adhesin-receptor binding or the action of chlamydial effectors. Experiments studying invasion in galectin-knockout cells are therefore warranted, in order to assess the relative importance of this interaction to chlamydial invasion generally, and its potential role in actin recruitment specifically.

### 2.2. Actin Modulation during Irreversible Chlamydial Attachment

Irreversible, temperature-sensitive attachment of *Chlamydia* to the host is the result of bacterial adhesin binding to specific host receptors. Indeed, the study of chlamydial invasion has identified an extensive portfolio of host and bacterial proteins that promote attachment (reviewed in more detail by Romero et al. [62]). Of these, several host receptors bound by *Chlamydiae* promote actin recruitment and endocytosis elsewhere, suggesting their possible role in the pathogen’s modulation of the actin cytoskeleton (Figure 1). For example, *C. trachomatis* serovar E and *C. muridarum* have been shown to directly bind the fibroblast growth factor (FGF), enabling molecular bridge formation between EBs and the host FGF receptor (FGFR) [63]. Intriguingly, the engagement of FGF-bound EBs with FGFR activated downstream signaling: phosphorylation of the FGFR substrate and docking protein FRS2α was increased by infection, and phosphorylated FRS2α was recruited to sites of EB attachment [63]. Typical binding of FGFR to its ligand results in internalization of the ligand–receptor complex, via Src/Eps8-dependent, clathrin-mediated endocytosis [64]. The involvement of Src is particularly significant, given that the kinase has been implicated in the recruitment of Rac1, WAVE, and Arp2/3 by the chlamydial invasion effector TarP (described below). Taken together, these observations suggest that *Chlamydia*-mediated activation of FGFR may aid in the recruitment of Src to the site of invasion.

An RNA interference screen of host factors contributing to invasion by *C. muridarum* identified the beta-isoform of the platelet-derived growth factor receptor (PDGFRβ) as another target of chlamydial attachment [33]. Much like FGFR, PDGFRβ is activated by EB binding and thereby promotes the tyrosine phosphorylation of multiple actin-modulatory proteins previously implicated in chlamydial invasion, including the nucleation promoting factors (NPFs) WAVE2 and cortactin, the Rac1 activator Vav2, and the chlamydial invasion effector TarP [33]. However, PDGFRβ activation was found to be dispensable for invasion, due to a compensatory mechanism dependent on activation of the Abelson (Abl) family of kinases [33]. This result suggests that PDGFRβ and Abl are elements of complementary invasion pathways, likely acting through a TarP-dependent mechanism given previous data linking TarP function to Vav2 and Abl (see below). Ultimately, a more thorough understanding of underlying mechanisms of each pathway is required, of which an essential first step is the identification of chlamydial factors that bind PDGFRβ or activate Abl kinases. Recent advances in *Chlamydia* transposon mutant strains may enable further study in this area, as would the use of Abl/PDGFRβ-deleted host cell lines generated via CRISPR/Cas9. Regardless, the substantial variety of host receptors with which invading EBs can interact suggests that *Chlamydiae* rely upon multiple host signaling pathways to mediate invasion; given the non-phagocytic nature of host cells, chlamydial modulation of those pathways thus maximizes the invasion efficiency. Ongoing study in organoid and in vivo models of *C. trachomatis* infection may better illustrate the relative contribution of each pathway to chlamydial invasion. 

### 2.3. TarP, a Multifunctional Actin-Recruiting Effector

As noted previously, the invasion of host cells by EBs occurs concomitantly with extensive tyrosine phosphorylation of proteins at the site of EB attachment [65,66,67]. Pulldown of these species by tyrosine phosphoantibodies and subsequent characterization revealed the presence of a bacterial effector, now known as TarP (translocated actin recruitment protein) [67]. TarP has since been shown to be translocated into the host cytosol by the chlamydial T3SS, where it has a multifunctional role in facilitating actin polymerization and subsequent internalization of the pathogen. TarP has been repeatedly implicated in the invasion of *C. trachomatis* and is highly conserved amongst all *Chlamydia* species with sequenced genomes [34,68], emphasizing a critical role for this effector in chlamydial pathogenesis.

The structure and functional domains of TarP can be divided into C- and N-terminal halves (Figure 2). The C-terminal domains are highly conserved amongst *C. trachomatis* serovars and other *Chlamydiae* [34], containing a proline-rich domain (PRD), an actin-binding domain (ABD), a leucine-aspartate (LD) motif that binds focal adhesion kinase (FAK), and a vinculin-binding domain (VBD). At sites of EB attachment and engagement of the chlamydial T3SS, the TarP PRD allows translocated TarP molecules to oligomerize [69]. This is presumed to create a high local concentration of ABD-bound actin at the invasion site, likely facilitating the formation of a trimeric actin nucleus for subsequent F-actin polymerization (Figure 3A) [69]. By contrast, the LD and VBD domains of TarP stimulate actin polymerization indirectly. Recruitment of the signaling kinase FAK by the TarP LD induces actin remodeling in an Arp2/3-dependent fashion (Figure 3B) [70]. Binding of the VBD to the actin adaptor protein vinculin promotes F-actin recruitment, which leads to further actin polymerization via an uncharacterized mechanism (Figure 3C) [71]. 

Intriguingly, the complex of host and chlamydial proteins recruited at the invasion site bears substantial similarity to a focal adhesion—A membrane-associated protein complex that couples the actin cytoskeleton to the extracellular matrix (ECM). Focal adhesions are assembled in an FAK- and Src-dependent fashion around the cytosolic domain of ECM-bound integrins [72,73,74,75,76], subsequently recruiting F-actin stress fibers to play an important role in host cell motility and signaling [77]. The resemblance of the chlamydial invasion complex to the focal adhesion is supported by a number of additional observations: the focal adhesion component paxillin is also recruited at sites of EB invasion, the invasion efficiency is markedly decreased during *C. caviae* infection of FAK- or vinculin-knockout MEFs, and an interaction between the chlamydial surface protein Ctad1 and host integrin-β1 supports the adhesion of *C. trachomatis* serovar E to host cells in vitro [71,78]. Given the predominantly basolateral localization of integrins in the polarized epithelial cells *Chlamydiae* typically infect, access to integrin-β1 receptors would require the disruption of intercellular junctions via micro-abrasions in the epithelium or loss of apicobasolateral polarity, as occurs during documented the epithelial-to-mesenchymal transition (EMT) of *Chlamydia*-infected epithelial cells [79]. The functional consequences of the recruitment of a focal adhesion-like structure to the site of chlamydial invasion of polarized cells are largely unexplored. Additionally, given that TarP expression is detectable between 8 and 18 h post-infection (hpi)—Long after invasion, but before the differentiation of infectious EBs—The possibility that this effector may have a post-invasion role modulating FAK- or vinculin-containing structures in the host is difficult to ignore [80]. 

The N-terminal half of *C. trachomatis* TarP contains between one and twelve copies of a tyrosine phosphodomain, with the copy number varying by serovar (Figure 2) [34,67,68,81,82]. This domain exhibits a high sequence similarity to targets of Src- and Abl-family kinases; accordingly, p60-Src, Yes, Fyn, and Abl are capable of phosphorylating TarP in vitro [83]. A variety of host proteins have been shown to interact with the TarP N-terminus upon its phosphorylation, including two guanidine exchange factors (Sos1 and Vav2) that enhance the activity of the GTPase Rac1 [84]. Rac1 activation by TarP subsequently initiates actin branching and polymerization, via the Rac1- and WAVE-dependent recruitment of an Arp2/3 complex (Figure 3D) [84,85]. Additionally, it has been demonstrated that the recruitment of Arp2/3 may enhance the actin-nucleating function of TarP’s C-terminal domains: the same in vitro pyrene-actin assay that established this function of TarP exhibited enhanced actin polymerization when Arp2/3 complexes were added [69]. The prevailing interpretation of this result is that the PRD/ABD domains and phosphodomains of TarP act in concert: the PRD and ABD domains promote the formation and extension of an initial (mother) actin filament, while the phosphodomains recruit Arp2/3 to facilitate branching from the mother filament.

The study of TarP has generally focused upon the *C. trachomatis* serovar homologs of the protein; perhaps unsurprisingly, characterization of the TarP orthologs has substantially complicated our understanding of the effector’s various functions. *C. muridarum* and *C. caviae* possess TarP orthologs absent of any phosphodomains [34], suggesting that TarP phosphorylation is largely dispensable for the invasion of mouse and guinea pig epithelial cells. Additionally, recent functional characterization of the *C. pneumoniae* TarP homolog CPn0572 has revealed an alternative actin-modulating function: CPn0572 expressed in HEK293T cells exhibited an alternative localization pattern to that of TarP, associating with F-actin filaments projecting from the actin-rich aggregates where TarP is typically observed [31]. When ectopically expressed in yeast, CPn0572 promotes the increased incidence of short F-actin bundles [30,31], which may be reminiscent of the F-actin bundles formed by *C. trachomatis* TarP [86]. This effect is dependent on the CPn0572 VBD, as well as a novel domain that binds F-actin (in contrast to the G-actin-binding TarP/CPn0572 ABD) [30]. Furthermore, CPn0572 binding displaces the actin depolymerizing factor cofilin, suggesting that CPn0572 promotes the formation of actin bundles indirectly by inhibiting their disassembly [31]. The purpose of these structures to *C. pneumoniae* invasion is unclear, though cofilin and cofilin-displacing effectors have been implicated in the invasion of other intracellular pathogens [87,88].

The TarP orthologs of *C. pneumoniae*, *C. muridarum*, and *C. caviae* have the potential to provide critical insight into how requirements for the induction of phagocytosis differ, depending on the host tissue type (genital/ocular vs. pulmonary epithelium) or host organism (human vs. murine). Heterologous complementation of virulence factors has been used to interrogate the mechanism of host tropism for other pathogens. However, the relative intractability of *Chlamydia* to genetic manipulation generally and TarP knockout specifically has hitherto precluded this approach [89]. The recent development of *Chlamydia* conditional expression/knockout strains may enable the use of this method [90,91], thereby illustrating the relative contribution of TarP’s many actin-modulating functions to invasion.

While the field’s understanding of TarP’s functional domains is extensive, the mechanism by which TarP translocation is regulated by pathogen adhesion to host cells remains somewhat unclear, beyond a presumed dependence on the activity of the chlamydial type III secretion system. In the Gram-negative pathogen *Yersinia pestis*, activation of the type III secretion system is cell contact-dependent, and possibly mediated by the YopN-TyeA-YscB-SycN complex—termed the calcium plug due to the complex’s calcium-dependent restriction of effector secretion in a cell-free context [92]. Only one complex member has been identified in *Chlamydia*: CopN (homologous to YopN) [93]. However, the chlamydial T3SS exhibits a similar sensitivity to calcium-mediated inhibition—indeed, Jamison and Hackstadt observed that Ca^2+^ chelation by EGTA was required for cell-free chlamydial effector translocation to occur [94]. Cell-free TarP translocation by chlamydial EBs is demonstrably induced by cholesterol and sphingomyelin-enriched liposomes [94], which suggests a contact-dependent model of T3SS activation dependent on specific lipids. We refer the reader to reviews from Betts-Hampikian, Ferrell, and Fields for a comprehensive discussion of the chlamydial T3SS [95,96].

The field’s current understanding of TarP’s various functions indicates that this effector serves as a signaling scaffold, targeting host actin-remodeling machinery to the site of chlamydial invasion. The potential of TarP to nucleate actin likely serves to enhance the actin-dependent invasion of non-phagocytic cell types. It is possible that invasion is mechanistically linked with the nascent *Chlamydia*-containing vacuole’s subsequent evasion of fusion with bactericidal host lysosomes, as has been observed with other intracellular pathogens. For example, recruitment of the small GTPase Rab5 to the vesicular surface during infection by *Brucella abortus* has been implicated in trafficking of the *Brucella* to a hospitable intracellular compartment; critically, Rab5 recruitment has been observed concomitant with invasion, during invagination of the clathrin-coated *Brucella*-containing vesicle [97]. The action of TarP and other actin-modulating effectors may similarly promote trafficking to a pathogen-favorable environment—elucidating the mechanistic connection between the invasion and chlamydial modulation of endocytic machinery is thus an intriguing topic for further study.

### 2.4. Actin-Depolymerizing Chlamydial Effectors

It is important to note that the induction of actin polymerization and bundling is not a terminal state for chlamydial invasion. After restructuring the cytoskeleton to initiate endocytosis, it is then necessary to restore steady-state actin dynamics in the host for invasion to proceed. Multiple candidates for this function have been identified in *Chlamydiae*—one such example is the *C. trachomatis* protein TmeA, a T3SS effector shown to associate with host AHNAK by a yeast two-hybrid assay [98,99]. AHNAK is a ubiquitous phosphoprotein implicated in cortical actin maintenance [100]; specifically, a C-terminal fragment of the protein has been shown to induce F-actin bundling [101]. Accordingly, the ectopic expression of TmeA in HeLa cells produced striking morphological changes consistent with the redistribution of actin [98]. This implies that TmeA may relieve the actin-bundling effects of TarP after invasion has begun; however, further study has complicated our understanding of TmeA’s true role in pathogenesis.

A recent report by McKuen et al. adds more nuance to the relationship between TmeA and AHNAK [102]. While TmeA can recruit AHNAK to the site of invasion and inhibit AHNAK-mediated actin bundling in vitro, *C. trachomatis* invasion of AHNAK-knockout mouse embryonic fibroblasts (MEFs) had no invasion defect relative to that of wild-type cells [102]. Combined with the reported AHNAK-independent invasion defect in a TmeA-knockout *C. trachomatis* strain, these results suggest that TmeA facilitates invasion via an unknown, AHNAK-independent mechanism [102]. That being the case, how does TmeA-mediated inhibition AHNAK affect the host and benefit the pathogen? One possibility is that TmeA possesses an AHNAK-dependent, post-invasion function. However, the lack of an observed defect in replication or EB recovery in AHNAK-knockout MEFs would seem to contradict this hypothesis [102]. It is possible that any effect of AHNAK deficiency on pathogenesis was obscured by the choice of host; a murine, mesenchymal model for *C. trachomatis* infection likely has differing requirements for invasion relative to the polarized genital/ocular epithelium the pathogen infects in vivo. Unfortunately, the total knockdown of AHNAK has proven technically challenging in many cell lines [100,103]; continuing advances in CRISPR/Cas9 and related gene-editing techniques may thus be required to address this lingering question in a more physiologically relevant model.

The chlamydial toxin CT166 has also been proposed to indirectly mediate actin depolymerization, acting as an inhibitor of host actin-polymerizing factors. CT166 contains a DXD amino acid motif with homology to the *Clostridium difficile* toxin TcdB, a glucosylator of the Rho-family GTPases Rac1 and Cdc42 (both well-known regulators of actin polymerization) [104,105]. Glucosylation of Rac1/Cdc42 by TcdB is inhibitory and irreversible, occurring at a catalytic threonine residue essential for GTP binding [105]. The downstream effect of TcdB activity is a dramatic redistribution of filamentous actin, resulting in cell shrinking, the loss of stress fibers, and host cell death [105,106].

CT166 is highly variable between *C. trachomatis* serovars: serovar D possesses a severely truncated CT166 that nevertheless retains the TcdB-homologous glucosylating domain, whereas serovar L2 lacks the gene entirely [104]. The inoculation of HeLa cells with *C. trachomatis* serovar D or *C. muridarum* (which possesses a full-length copy of CT166) at a high multiplicity of infection resulted in morphological defects and cytotoxicity, reminiscent of the effects of TcdB [104]. Importantly, cytotoxicity was not observed in a comparable infection using the CT166-deficient *C. trachomatis* serovar L2, suggesting that this phenomenon was CT166-specific [104]. It has since been shown that CT166 expressed in HeLa cells inactivates the Rac/Cdc42 relative Ras, and that this effect could be ablated via mutation of the DXD domain [107,108].

CT166’s role during invasion is somewhat unclear, given that the CT166-deficient *C. trachomatis* serovars can invade host cells in vitro with a comparable efficiency to those that possess the effector [104]. As with TmeA, this may be a product of the choice of host model; it remains to be seen whether CT166’s glucosylating activity is equally dispensable in three-dimensional or organismal models of chlamydial infection. Given that the proposed function of CT166 runs counter to chlamydial invasion’s established dependence on actin polymerization, it follows that the timing of CT166 delivery and activation is tightly regulated, in order to ensure that the toxin’s activity is beneficial to the pathogen. Further study of CT166 may therefore inform the timing and kinetics of chlamydial effector delivery, as well as provide insight into the regulation of actin dynamics by the host.

## 3. Life after Invasion: *Chlamydia* Modulates Actin for Inclusion Stability and Host Egress

Given the robust and multifactorial nature of cytoskeletal restructuring during chlamydial invasion, it should not be surprising that the pathogen continues to manipulate the cytoskeleton once internalized by the host. Initial study of inclusion development has shown that *Chlamydiae* restructure the microtubule network to foster the development of their replicative niche. Another early observation of infections in vitro was that the treatment of infected cells with the microtubule polymerization inhibitor colchicine produces a marked increase in inclusion size [109]. Furthermore, rapid and inclusion-proximate microtubule assembly has been observed after washout of the microtubule-disrupting agent nocodazole [110,111]. It has been shown that *Chlamydia*-directed microtubule restructuring traffics the nascent *C. trachomatis* inclusion to the microtubule-organizing center (MTOC)—an outcome that is presumed to enable the chlamydial scavenging of nutrients from the host [111,112]. Indeed, the inclusion demonstrably interacts with a variety of nutrient-rich, MTOC-associated organelles, including the ER and multivesicular bodies (MVBs) [113,114,115,116,117]; pharmacological disruption of inclusion-ER/MVB association impairs chlamydial growth [113,115], consistent with a model of the chlamydial theft of nutrients from these organelles.

The role of actin in post-invasion chlamydial pathogenesis is more poorly characterized. In 1989, Campbell et al. discovered that mature inclusions are surrounded by a cage of actin and intermediate filaments [112], but it was only in the past decade that a mechanistic understanding of this structure’s assembly and importance became clear. Additionally, recent study has shown that actin and actin-related structures play a role in pathogen egress, by facilitating the extrusion of intact chlamydial inclusions from the host.

### 3.1. Actin-Mediated Reinforcement of the Inclusion

As noted previously, early live-cell imaging experiments established the rapid recruitment of actin at the site of chlamydial invasion of the host [40]. Critically, the actin-rich structures surrounding invasion sites are transient, meaning that the actin-rich cage observed surrounding nascent inclusions is a distinct structure assembled later, likely with a separate function [112,118,119]. It has since been demonstrated that actin affords considerable stability to the inclusion: strikingly, treatment with 1% Triton X-100 does not significantly alter the morphology of the inclusion, despite the near-total solubilization of its membrane [118]. This apparent resistance to nonionic detergents was subsequently shown to depend on an F-actin ring surrounding the inclusion—actin disruption via latrunculin-A/B compromised the inclusion membrane integrity, leading to the detection of bacterial lipopolysaccharide (LPS) and antibody-labeled *Chlamydia* in the cytosol of infected cells [118]. Importantly, microtubule disruption with nocodazole did not produce similar changes in inclusion morphology. Taken together, these results imply a specific role for actin in structurally reinforcing the maturing inclusion [118].

The mechanism of actin recruitment to the inclusion remains somewhat unclear. Actin-rich structures in the host are largely indistinguishable from uninfected cells, in stark contrast to the pathogen’s extensive remodeling of the microtubule network [110,112]. Furthermore, F-actin rings associated with maturing inclusions (30 hpi) are unaffected by the pharmacological disruption of stress fibers and cortical actin, including the inhibition of Rac1, Cdc42, ROCK, and myosin II [118]. Intriguingly, the inhibition of RhoA (either via siRNA knockdown or treatment with the *Clostridium botulinum* toxin C3-transferase) reduced the incidence of F-actin rings, and ablated the inclusion resistance to TX-100 [118]. Collectively, these results imply that the cage’s actin component is assembled independently of host actin-rich structures.

More recent study of actin recruitment to the inclusion has complicated the model of cage assembly. Live-cell imaging of actin recruitment to mature inclusions (44 and 68 hpi) demonstrates sensitivity to formin inhibition (but not Arp2/3) [119], suggesting that cage formation may indeed depend upon the de novo, unbranched polymerization of actin at inclusions. However, actin recruitment at these time points occurred independently of RhoA and was sensitive to myosin II inhibition—seemingly in direct contradiction to ring assembly dynamics at earlier stages of infection [118,119]. How might data from these two stages of infection be reconciled? One explanation is that initial assembly and subsequent maintenance of the cage’s actin component occurs via two distinct mechanisms—the former requiring RhoA, and the latter requiring formins and myosin II. A more longitudinal study of chlamydial actin recruitment is required to test this hypothesis, and may reveal how the mechanism of cage assembly changes with maturation of the inclusion.

Intermediate filaments have also been shown to contribute to the stability and function of the inclusion cage. While the infection of vimentin-knockout MEFs (a cell line also deficient in cytokeratins) produced F-actin rings comparable to the infection of wild-type cells, these rings lacked the highly compact and ordered morphology of their wild-type counterparts [118]. Inclusions in vimentin-knockout cells also lacked the previously observed resistance to TX-100. Taken together, these results suggest that the recruitment or assembly of F-actin rings surrounding the inclusion provides a scaffold for further inclusion reinforcement by intermediate filaments [118]. The chlamydial protease CPAF has been shown to cleave vimentin (as well as cytokeratin-8/18) within the protein’s head domain, partially inhibiting its ability to form filamentous structures [118,120,121]. It is postulated that this interaction permits highly dynamic maintenance of the actin/filament cage, allowing the structure to accommodate an ever-expanding inclusion.

Collectively, these findings suggest that *Chlamydiae* dynamically reinforce the developing inclusion with F-actin and intermediate filaments (Figure 4). While the precise mechanism of F-actin synthesis and recruitment to the inclusion is somewhat unclear, the interconnected nature of the cytoskeleton would suggest that pathogen-directed actin restructuring may affect other cytoskeletal components, like the microtubule network. Given the established importance of the MTOC and vesicular trafficking to chlamydial growth (reviewed in detail by Nogueira et al. [122]), disruption of the microtubule network by any means has significant implications for the pathogen. Therefore, further study of the actin cage seems warranted, in order to evaluate a possible role for this structure in the chlamydial modulation of microtubule dynamics.

### 3.2. The Role of Actin in Chlamydial Extrusion

Like other intracellular pathogens, *Chlamydiae* can exit the host via the induction of host cell lysis. However, another, less destructive, pathway exists as well. Early indication of an alternative mechanism of egress came in the observation of “scarred” host cells that—while intact—lacked a substantial portion of their plasma membranes [123]. Hybiske and Stephens proved the existence of non-lytic egress by taking advantage of the tendency of the chlamydial inclusion to exclude host cytosolic proteins [61]. Upon infecting GFP-expressing HeLa cells with *C. trachomatis*, they determined that while half of the observed inclusions became GFP-permeable during host cell lysis, the remaining half fused with the plasma membrane, excluding GFP throughout egress from the host, suggesting that an exocytotic event had occurred [61]. The incidence of this event (termed extrusion) was sensitive to disruption of the actin cytoskeleton by latrunculin B; intriguingly, the same did not hold true for nocodazole treatment, indicating that extrusion occurs independent of the microtubule network and conventional vesicular trafficking [61]. Further inhibitor treatments revealed a dependence on the nucleation promoting factor N-WASP, Rho-family GTPases, and myosin II. Taken together, these data imply an extrusion mechanism reliant on F-actin polymerization and bundling [61].

In 2013, Lutter et al. observed the recruitment of a complex of myosin-related host proteins to the inclusion surface, including myosin IIa; myosin IIb; and the phosphorylated, active forms of myosin light chain 2 (MLC2) and myosin light chain kinase (MLCK) [124,125]. The depletion of these myosin-activating factors via siRNA knockdown reduced the incidence of extrusion events, suggesting that *Chlamydia*-directed activation of myosin II promotes non-lytic egress of the pathogen [124]. Accordingly, the inclusion surface protein CT228 has been shown to directly interact with MYPT1, a negative regulator of myosin II activity. MYPT1 acts as a regulatory subunit of the phosphatase PP2—upon binding, MYPT1 alters the phosphatase’s binding specificity, resulting in the PP2-mediated dephosphorylation (and, thereby, inactivation) of MLC2 [126]. The binding of MYPT1 to PP2 can be inhibited by the former’s phosphorylation—given that CT228 preferentially recruits phosphorylated MYPT1 to the inclusion surface, it was initially proposed that CT228 might facilitate extrusion via MYPT1 inhibition [124].

However, the recent finding that the TargeTron-mediated knockout of CT228 increases the extrusion incidence (consistent with the siRNA-mediated knockdown of MYPT1) indicates an alternative, extrusion-inhibitory role for CT228 [127]. Combined with the earlier observation of the robust dephosphorylation of MYPT1 late in infection (42 hpi), this result suggests that CT228-mediated MYPT1 recruitment may result in MYPT1 activation instead [127], perhaps serving as a mechanism to regulate myosin II activity at the inclusion (and thereby regulating extrusion-mediated egress of the pathogen). Extrusion is further regulated by the interaction of the chlamydial inclusion membrane protein MrcA with IPTR3, an inositol 1,4,5-triphosphate receptor that acts as a Ca^2+^ channel [128]. Accordingly, the regulatory action of this complex further required the host cell Ca2+ sensor STIM1 [128]. It is important to note that premature host egress (i.e. prior to the differentiation of *Chlamydia* into infectious EBs) is a disastrous outcome for the pathogen that precludes subsequent infection of the surrounding tissue. The temporal regulation of MYPT1 activity via the expression of CT228 may therefore constitute a means by which *Chlamydiae* inhibit extrusion until differentiation is complete, thereby ensuring the release of invasion-competent organisms.

Further study has put forward a connection between the inclusion’s actin cage and extrusion, specifically involving the septin family of GTP-binding proteins [129]. The function of septins is primarily structural—upon binding to GTP, septins oligomerize to form filamentous structures that associate with F-actin and other cytoskeletal components [130]. Accordingly, septin-2, 9, and 11 were recently observed to colocalize with the actin-rich structures surrounding the maturing inclusion [129]. Septin-9 knockdown both ablated actin recruitment to the inclusion and resulted in a 2 to 3-fold reduction of extrusion events [129], suggesting that these processes are functionally interrelated. This exciting result prompts several questions: does the structural content of the inclusion cage bias chlamydial egress via one pathway over another? How might the various structural components of the cage contribute to the activity of CT228 and myosin II? Further research in this area may indicate how actin, intermediate filaments, and septins contribute to vesicular trafficking and exocytosis, as well as provide insight into their specific role in chlamydial egress.

## 4. Conclusions and Summary

While mechanistic study of the actin modulation by *Chlamydiae* is both robust and ongoing, it is important to acknowledge that actin remodeling does not occur in a vacuum, and that the effect of *Chlamydia*-restructured actin on the host cell has been largely unassessed. The actin cytoskeleton is much more than a structural network for the host—in addition to driving endocytosis and cellular motility, mounting evidence implicates actin dynamics in transcriptional regulation.

The tensile force transmitted across actin filaments has been demonstrated to induce changes in gene expression, via the activity of actin- and/or tension-responsive signaling pathways. For example, the serum response factor (SRF) is demonstrably activated by regulators of actin dynamics, such as the cofilin-inhibiting LIM kinase-1 [131]. The application of tensile force to cardiac fibroblasts in vitro has shown the tension-dependent expression of α-smooth muscle actin, an SRF regulatory target [132]. Dysregulation of SRF activity has been associated with induction of the epithelial-to-mesenchymal transition (EMT)—a process through which epithelial cells transdifferentiate into scar-forming fibroblasts [133,134]. Intriguingly, *Chlamydia* infection has recently been shown to induce EMT, which likely promotes the scar-forming pathology observed in chronic *C. trachomatis* infections [79,135,136,137]. Could chlamydial restructuring of actin contribute to this effect? Further characterization of actin-regulated gene expression in *Chlamydia*-infected cells appears warranted.

Force transduction across the actin cytoskeleton also has demonstrable effects on the nuclear architecture. The inner nuclear membrane (INM) is reinforced by a network of intermediate filaments and associated proteins—including lamin-A/C and emerin—that are structurally integrated with the cytoskeleton via the LINC (linker of nucleoskeleton and cytoskeleton) complex [138]. Tension-mediated phosphorylation of emerin is associated with redistribution of the LINC complex and nuclear actin, which in turn are respectively associated with chromatin remodeling and myocardin-related transcription factor (MRTF) activity [139,140,141,142]. Given the multifaceted way in which *Chlamydiae* restructure actin and modulate the activity of myosin II—a contractile, actin-crosslinking protein—it is plausible that infection has the secondary effect of modulating tension- and actin-responsive gene expression, and that characterization of this effect may provide insight into chlamydial pathogenesis. 

The polarized epithelial mucosa that *Chlamydiae* typically infect often react to bacterial insult via the concerted exfoliation and apoptosis of infected cells—a process that requires the degradation of focal adhesions and related cell-ECM adherent structures, as well as extensive remodeling of the actin cytoskeleton [143]. Exfoliation and apoptosis of the host cell is clearly an unfavorable outcome for any intracellular pathogen. Accordingly, there is evidence indicating that other intracellular bacteria inhibit this host response to infection. The *Shigella flexneri* effector OspE has been shown to reinforce sites of cell-ECM contact through an interaction with integrin-linked kinase (ILK), which results in an increased incidence of integrin-β1 on the host cell surface and the consequent assembly of focal adhesions [144]. A secondary effect of the OspE modulation of ILK is the stabilization of focal adhesions, via the inhibition of focal adhesion kinase (FAK). Given that the chlamydial effector TarP is known to recruit FAK (albeit in the context of invasion) [70], it is tempting to speculate that *Chlamydia* might inhibit exfoliation in a similar fashion. Ultimately, further characterization of host cell adherence and exfoliation during chlamydial infection is necessary to address this hypothesis.

In summary, ongoing study of actin modulation by *Chlamydiae* has revealed the multifaceted way in which these pathogens restructure the host actin cytoskeleton. During the two-stage attachment of chlamydial EBs to the host cell surface, the pathogen engages with a variety of host surface proteins—including galectin-1, FGFR, and PDGFRβ—that in turn facilitate the recruitment of signaling factors promoting actin polymerization at the attachment site [33,52,53,54,63]. The subsequent delivery of chlamydial effectors by the pathogen’s type III secretion system then induces actin-dependent invasion of the host cell. The chlamydial effector TarP contributes to this effect in multiple ways: via the recruitment of F-actin and Arp2/3 actin-polymerizing complexes, via direct nucleation of actin polymerization, and by providing a scaffold for actin-modulatory signaling [69,70,71,84,85]. While the mechanism by which actin regulation at the invasion site returns to a steady state remains somewhat unclear, the chlamydial effectors TmeA and CT166 have been demonstrated to inhibit F-actin bundling and polymerization, respectively, and may contribute to this process [98,99,102,104,107,108]. Once invasion is complete, actin is recruited in either an RhoA- or ROCK-dependent fashion to the maturing inclusion alongside intermediate filaments and septins, providing dynamic structural reinforcement to *Chlamydia’s* replicative niche [118,119]. Finally, non-lytic chlamydial egress from the host depends on actin polymerization, as well as the modulation of myosin II by the inclusion membrane protein CT228 [61,124,127]. While the mechanistic underpinnings of actin restructuring by *Chlamydiae* are coming into focus, the effects of this restructuring on the host are relatively uncharacterized, presenting intriguing opportunities for future study.

## Figures and Tables

**Figure 1 ijms-21-00090-f001:**
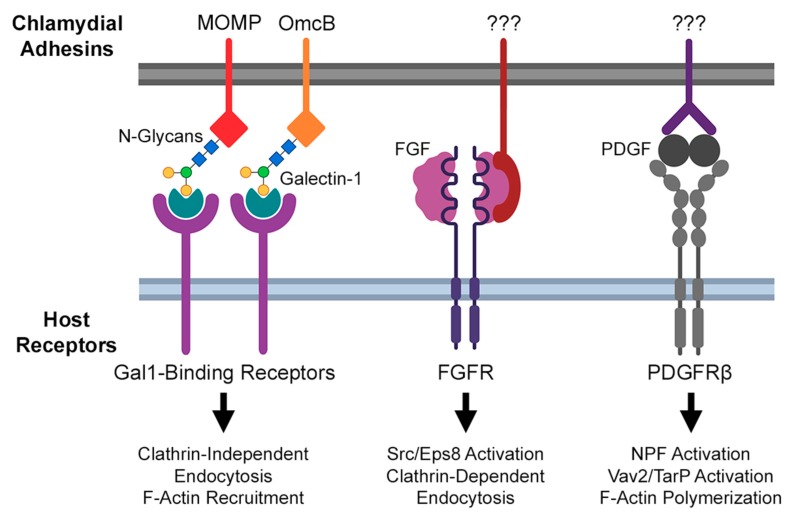
Summary of known chlamydial adhesin/host receptor interactions implicated in *Chlamydia*-directed modulation of actin.

**Figure 2 ijms-21-00090-f002:**
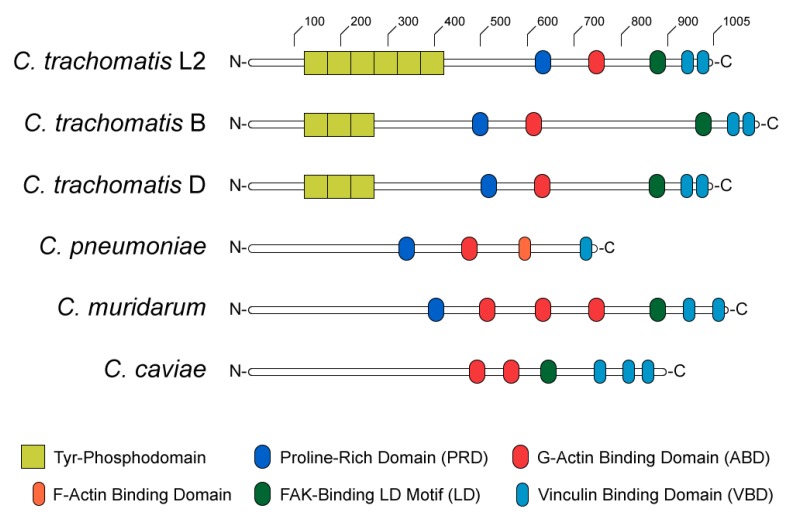
Diagram of TarP homologs in anogenital (L2), ocular (B), and genital (D) serovars of *Chlamydia trachomatis*, as well as *C. pneumoniae*, *C. muridarum*, and *C. caviae*.

**Figure 3 ijms-21-00090-f003:**
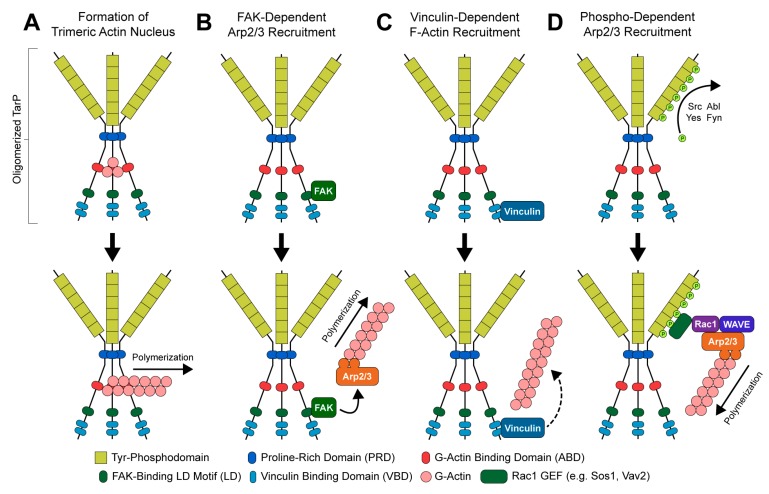
Mechanisms of TarP-directed actin modulation: (**A**) oligomerization-dependent formation of trimeric actin nucleus; (**B**) focal adhesion kinase (FAK)-dependent recruitment of the Arp2/3 complex; (**C**) vinculin-dependent recruitment of F-actin; (**D**) phosphorylation-dependent recruitment of an Arp2/3-activating protein complex.

**Figure 4 ijms-21-00090-f004:**
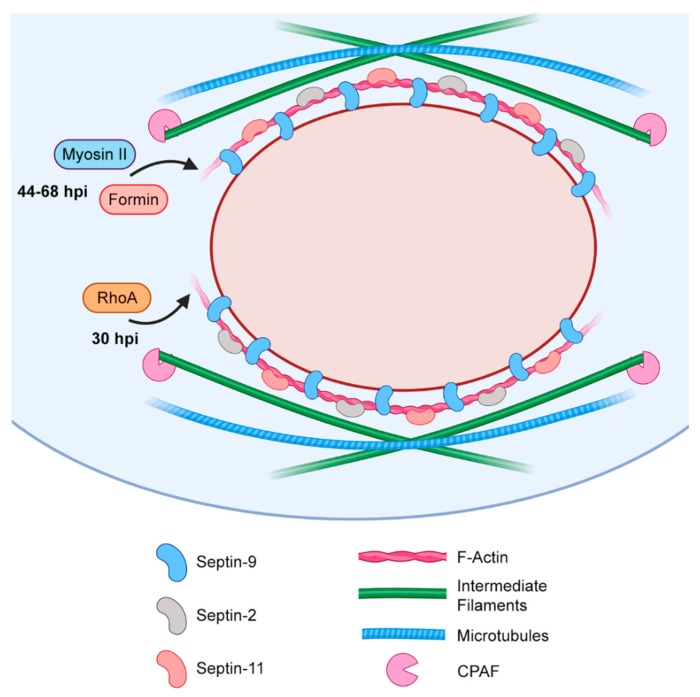
Diagram of the filamentous cage surrounding the maturing chlamydial inclusion.

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
