# Peer review of "Pathogenic Puppetry: Manipulation of the Host Actin Cytoskeleton by Chlamydia trachomatis"

_ijms, 2019, doi:10.3390/ijms21010090_

Round 1

Reviewer 1 Report

This is a well written and well organized review.

In the section reviewing Tarp domains, the correlation of tyrosine rich repeats with clinical biovars has been previously published and should be included in the referencing (PMID: 20605986) [lines 193-201; line 233].

Line 76: First use of C. pneumonaie - expand out genus

Line 77: C. muridarum and C. caviae are being used for the first time. Please expand the genus same.

Line 221 - Chlamydia is not italicized 

When reviewing the role of TarP in F-actin polymerization, add in the reference by Ghosh et al., 2018 (PMID:  29660331)   

 A recent study by Nguyen et al, 2018 ( PMID: 29543918) also implicates ca2+ and ITPR3 as regulating extrusion events and MLC2 phosphorylation.  Please add into the discussion somewhere is lines 427-449

EMT is defined twice: Lines 474 and 221. Only once is needed.

After line 530 in abbreviations: please add  MRTF, ECM, INM, ER, FGF, FGFR

Overall edits: sometimes "Chlamydial" is capitalized and sometimes it is not (example:  lines 271 and 416). Please edit to be consistent.

Author Response

This is a well written and well organized review.

We thank the Reviewer’s encouraging comment.

In the section reviewing Tarp domains, the correlation of tyrosine rich repeats with clinical biovars has been previously published and should be included in the referencing (PMID: 20605986) [lines 193-201; line 233].

The reference has been added at appropriate places throughout the manuscript.

Line 76: First use of C. pneumonaie - expand out genus

This has been corrected.

Line 77: C. muridarum and C. caviae are being used for the first time. Please expand the genus same.

These have been corrected.

Line 221 - Chlamydia is not italicized

This has been corrected.

When reviewing the role of TarP in F-actin polymerization, add in the reference by Ghosh et al., 2018 (PMID:  29660331).

The following phrase has been added (line 255-256; numbering in the new version of the manuscript). “, which may be reminiscent of the F-actin bundles formed by C. trachomatis TarP (Ghosh reference)”.

A recent study by Nguyen et al, 2018 ( PMID: 29543918) also implicates ca2+ and ITPR3 as regulating extrusion events and MLC2 phosphorylation.  Please add into the discussion somewhere is lines 427-449.

The following sentences have been added (Lines 462-465).

Extrusion is further regulated by the chlamydial protein MrcA interaction with the cellular protein Ca2+ channel inositol-1,4,5-trisphosphate receptor, type 3 (ITPR3). The action of this complex further required the host cell Ca2+ sensor STIM1 [Nguyen reference].

EMT is defined twice: Lines 474 and 221. Only once is needed.

This has been corrected.

After line 530 in abbreviations: please add  MRTF, ECM, INM, ER, FGF, FGFR

These abbreviations have been included in the list.

Overall edits: sometimes "Chlamydial" is capitalized and sometimes it is not (example:  lines 271 and 416). Please edit to be consistent.

These have been corrected.

Reviewer 2 Report

This is an extensive review of the role of host actin in all stages of Chlamydia infection of host cells - including adhesion, entry, vacuolar maintenance and extrusion.  A particular focus is on potential mechanisms by which Chlamydia temporally control actin polymerization and depolymerization.  

The authors take care to indicate needed future studies in each section of the review.  This includes potential studies that take advantage of newly developed technologies (e.g. Tn-based mutagenesis of Chlamydia and CRISPR/Cas9 knockouts in host cells).

This is a very well-written review that was informative and provided an extensive survey of the literature.  Relevant questions were raised throughout the review and speculation based on the existing literature was provided appropriately.  This review is informative to the nonexpert and will be of use to experts in the Chlamydia field.

The only minor suggestion I would make is that it would be helpful to include a brief discussion of the type 3 secretion system in section 2.3.  This discussion might include what is know about the Chlamydia T3SS, especially with regard to recognition of host cell contact and what about this contact activates the translocation process.  

Author Response

The only minor suggestion I would make is that it would be helpful to include a brief discussion of the type 3 secretion system in section 2.3.  This discussion might include what is known about the Chlamydia T3SS, especially with regard to recognition of host cell contact and what about this contact activates the translocation process. 

A brief section on how type III secretion might be triggered by cell contact has been added. The paragraph includes comparison with the contact-dependent and calcium regulated secretion by the Yersinia type III apparatus. For additional details on type III secretion in Chlamydia, the reader is referred to three comprehensive review articles.

The following paragraph has been added with the corresponding references (lines 271-284).

While the field’s understanding of TarP’s functional domains is extensive, the mechanism by which TarP translocation is regulated by pathogen adhesion to host cells remains somewhat unclear, beyond a presumed dependence on the activity of the chlamydial type III secretion system. In the Gram-negative pathogen Yersinia pestis, activation of the type III secretion system is cell contact-dependent, and possibly mediated by the YopN-TyeA-YscB-SycN complex – termed the calcium plug due to the complex’s calcium-dependent restriction of effector secretion in a cell-free context [92]. Only one complex member has been identified in Chlamydia, CopN (homologous to YopN) [93]. However, the chlamydial T3SS exhibits a similar sensitivity to calcium-mediated inhibition – indeed, Jamison and Hackstadt observed that Ca2+ chelation by EGTA was required for cell-free chlamydial effector translocation to occur [94]. Cell-free TarP translocation by chlamydial EBs is demonstrably induced by cholesterol and sphingomyelin-enriched liposomes [94], which suggests a contact-dependent model of T3SS activation dependent on specific lipids. We refer the reader to reviews from Betts-Hampikian, Ferrell, and Fields for a comprehensive discussion on the current understanding of the chlamydial T3SS [95,96].